# Endangered wild salmon infected by newly discovered viruses

**Gideon J Mordecai[1]\*, Kristina M Miller[2]\*, Emiliano Di Cicco[2,3], Angela D Schulze[2], Karia H Kaukinen[2], Tobi J Ming[2], Shaorong Li[2], Amy Tabata[2], Amy Teffer[4], David A Patterson[5], Hugh W Ferguson[6], Curtis A Suttle[1,7,8,9]\***

[1]Department of Earth, Ocean and Atmospheric Sciences, University of British Columbia, Vancouver, Canada; [2]Pacific Biological Station, Fisheries and Oceans Canada, Nanaimo, Canada; [3]Pacific Salmon Foundation, Vancouver, Canada; [4]Department of Forest Sciences, University of British Columbia, Vancouver, Canada; [5]Fisheries and Oceans Canada, Science Branch, Cooperative Resource Management Institute, School of Resource and Environmental Management, Simon Fraser University, Burnaby, Canada; [6]School of Veterinary Medicine, St. George's University, True Blue, Grenada; [7]Department of Microbiology and Immunology, University of British Columbia, Vancouver, Canada; [8]Department of Botany, University of British Columbia, Vancouver, Canada; [9]Institute for the Oceans and Fisheries, University of British Columbia, Vancouver, Canada

**\*For correspondence:**
gmordecai@eoas.ubc.ca (GJM);
Kristi.Saunders@dfo-mpo.gc.ca
(KMM);
suttle@science.ubc.ca (CAS)

**Competing interests:** The authors declare that no competing interests exist.

**Abstract** The collapse of iconic, keystone populations of sockeye (*Oncorhynchus nerka*) and Chinook (*Oncorhynchus tshawytscha*) salmon in the Northeast Pacific is of great concern. It is thought that infectious disease may contribute to declines, but little is known about viruses endemic to Pacific salmon. Metatranscriptomic sequencing and surveillance of dead and moribund cultured Chinook salmon revealed a novel arenavirus, reovirus and nidovirus. Sequencing revealed two different arenavirus variants which each infect wild Chinook and sockeye salmon. In situ hybridisation localised arenavirus mostly to blood cells. Population surveys of >6000 wild juvenile Chinook and sockeye salmon showed divergent distributions of viruses, implying different epidemiological processes. The discovery in dead and dying farmed salmon of previously unrecognised viruses that are also widely distributed in wild salmon, emphasizes the potential role that viral disease may play in the population dynamics of wild fish stocks, and the threat that these viruses may pose to aquaculture.
DOI: https://doi.org/10.7554/eLife.47615.001

## Introduction

Pacific salmon (*Oncorhynchus* spp.) species have supported coastal ecosystems and Indigenous populations surrounding the North Pacific Ocean for tens of millennia. Today, through their anadromous life history, salmon continue to transport nutrients between aquatic and terrestrial environments (*Cederholm et al., 1999*), supply the primary food sources for orca whales and sea lions (*Wasser et al., 2017*; *Willson and Halupka, 1995*; *Chasco et al., 2017*; *Thomas et al., 2017*) and provide economic livelihoods for local communities (*Noakes et al., 2002*). In the Northeast Pacific, widespread declines of Chinook (*O. tshawytscha*) and sockeye (*O. nerka*) salmon have occurred in the last 30 years, leading some populations to the brink of extirpation (*Peterman and Dorner, 2012*; *Heard et al., 2007*; *Miller et al., 2011*; *Jeffries et al., 2014*), and a cause of great concern to Indigenous groups, commercial and recreational fishers, and the general public. Although the exact number of salmon spawning in rivers is unknown, there are large declines in sockeye salmon over a

**eLife digest** Keystone species are animals and plants that play a pivotal role in supporting the ecosystems they live in, making their conservation a high priority. Chinook and sockeye salmon are two such species. These fish play a central role in the coastal ecosystems of the Northeast Pacific, where they have supported Indigenous populations for thousands of years.

The last three decades have seen large declines in populations of Chinook and sockeye salmon. One factor that may be involved in these declines is viral infection. In the last ten years, advances in DNA sequencing technologies have led to the discovery of many new viruses, and Mordecai et al. used these technologies to look for new viruses in Pacific salmon.

First, Mordecai et al. looked for viruses in dead and dying salmon from farms and discovered three previously unknown viruses. Next, they screened for these viruses in farmed salmon, hatchery salmon and wild salmon to determine their distribution. Two of the viruses were present in fish from the three sources, while one of the viruses was only found in farmed fish. The fact that the three viruses are distributed differently raises questions about how the viruses are transmitted within and between farmed, hatchery and wild salmon populations.

These findings will aid salmon-conservation efforts by informing the extent to which these viruses are present in wild salmon populations. Future work will focus on determining the risks these viruses pose to salmon health and investigating the potential for exchange between hatchery, farmed and wild salmon populations. While farmed Pacific salmon may pose some transmission risk to their wild counterparts, they also offer the opportunity to study disease processes that are not readily observable in wild salmon. In turn, such data can be used to develop policies to minimize the impact of these infectious agents and improve the survival of wild salmon populations.

DOI: https://doi.org/10.7554/eLife.47615.002

large geographic area (*Peterman and Dorner, 2012*). Similarly, Chinook salmon stocks are at only a small percentage of their historic levels, and more than 50 stocks are extinct (*Heard et al., 2007*).

It is thought that infectious disease may contribute to salmon declines (*Miller et al., 2011*), but little is known about infectious agents, especially viruses, endemic to Pacific salmon. Infectious disease has been identified as a potential factor in poor early marine survival in migratory salmon; an immune response to viruses has been associated with mortality in wild migratory smolts and adults (*Miller et al., 2011*; *Jeffries et al., 2014*), and in unspecified mortalities of salmon in marine net pens in British Columbia (BC) (*Miller et al., 2017*; *Di Cicco et al., 2018*). For instance, immune responses to viruses such as Infectious haematopoietic necrosis virus (IHNV) and potentially undiscovered viruses, have been associated with mortality in wild juvenile salmon (*Jeffries et al., 2014*). This is an important observation as mortality of juvenile salmon can be as high as ~90% transitioning from fresh water to the marine environment (*Clark et al., 2016*). Together, these suggest that there are undiscovered viruses which may contribute to decreased survival of Pacific salmon but a concerted effort to look for viruses that may contribute to mortality has been absent.

Here, virus-discovery was implemented to screen for viruses associated with mortality. Together, sequencing of dead or moribund aquaculture salmon and live-sampled wild salmon, in-situ hybridization, and epidemiological surveys revealed that previously unknown viruses, some of which are associated with disease, infect wild salmon from different populations.

## Results and discussion

Fish were screened against a viral disease detection biomarker panel (VDD) that elucidates a conserved transcriptional pattern indicative of an immune response to active RNA viral infection (*Miller et al., 2017*). For instance, in a previous study, we showed that 31% of moribund Atlantic salmon were in a viral disease state, and half of these were not known to be positive for any known RNA viruses (*Di Cicco et al., 2018*). Individuals that were strongly VDD-positive, but negative for any known salmon viruses (e.g. Piscine orthoreovirus, Erythrocytic necrosis virus, Infectious pancreatic necrosis virus, Infectious hematopoietic necrosis virus, Infectious salmon anaemia virus and Pacific salmon paramyxovirus) were subject to metatranscriptomic sequencing. The sequencing revealed viral transcripts belonging to members of the *Arenaviridae*, *Nidovirales* and *Reoviridae*,

three evolutionarily divergent groups of RNA viruses (*Figure 1*) that can be highly pathogenic

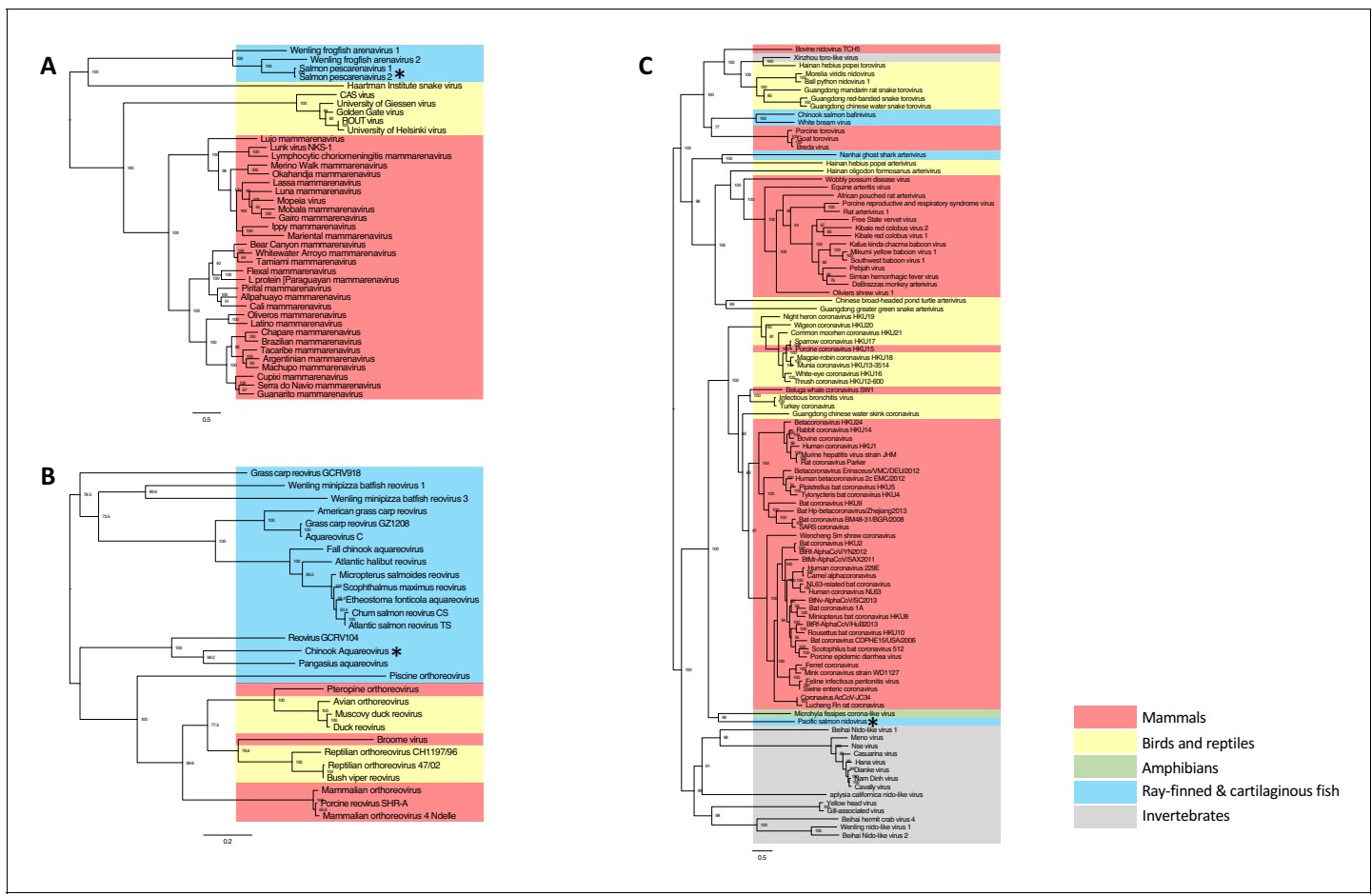

**Figure 1.** Maximum likelihood phylogenetic tree based on MAFFT alignments of the predicted replicase protein of (**A**) Salmon pescarenavirus and related arenaviruses, (**B**) Chinook Aquareovirus and related Aqua and orthoreoviruses and (**C**) Pacific salmon nidovirus and related Nidovirales. Sequences from this study are marked with an asterisk, scale bar represents the number of amino substitutions per site, node labels show the bootstrap support and host groups are shown by colour. Trees are mid-point rooted, so do not necessarily represent the ancestral relationship of the viruses. Amino acid alignments have been provided in the source data for *Figure 1*.

DOI: https://doi.org/10.7554/eLife.47615.003

The following source data and figure supplements are available for figure 1:

**Source data 1.** Arenavirus amino acid alignment.
DOI: https://doi.org/10.7554/eLife.47615.006

**Source data 2.** Nidovirus amino acid alignment.
DOI: https://doi.org/10.7554/eLife.47615.007

**Source data 3.** Reovirus amino acid alignment.
DOI: https://doi.org/10.7554/eLife.47615.008

**Source data 4.** Arenavirus phylogenetic tree.
DOI: https://doi.org/10.7554/eLife.47615.009

**Source data 5.** Reovirus phylogenetic tree.
DOI: https://doi.org/10.7554/eLife.47615.010

**Source data 6.** Nidovirus phylogenetic tree.
DOI: https://doi.org/10.7554/eLife.47615.011

**Figure supplement 1.** Genome organisation and coverage.
DOI: https://doi.org/10.7554/eLife.47615.004

**Figure supplement 1—source data 1.** Viral genomic nucleotide sequences.
DOI: https://doi.org/10.7554/eLife.47615.005

(*Yun and Walker, 2012*; *Liang et al., 2014*; *Weiss and Leibowitz, 2011*).

One of the challenges of viral discovery in fish is that the proportion of viral transcripts in vertebrate metatranscriptomic libraries is small compared to the number of transcripts from the host and other contaminating sequences (*Geoghegan et al., 2018*; *Zhang et al., 2019*). However, we were able to achieve near-coding complete genomes for the three new viruses (*Figure 1—figure supplement 1A and B*). The genomic organisation of the newly discovered viruses was consistent with related viruses in fish. For instance, SPAV has three genomic segments, as shown for other arenaviruses in fish (*Shi et al., 2018*). High-throughput RT-PCR screening of >6000 wild juvenile Chinook and sockeye salmon showed dissimilar geographical distributions of infected fish, reflecting differences in epidemiological patterns of transmission and infection dynamics for each of the viruses (*Figure 2*).

The distribution and abundance of the different viruses varied markedly. Arenaviruses were relatively common (*Figure 2—figure supplement 1*) and geographically widespread in migratory juvenile Chinook and sockeye salmon in the marine environment (*Figure 2*, *Figure 2—figure supplement 2*). Whereas, the nidovirus was spatially localised and predominantly observed at high prevalence over multiple years in Chinook salmon leaving freshwater hatcheries (*Figure 2*). Finally, the reovirus was detected only in farmed Chinook salmon (*Figure 2* and *Figure 2—figure supplement 1*).

With the exception of their relatively recent discovery in snakes (*Stenglein et al., 2012*) and frogfish (*Shi et al., 2018*), arenaviruses were thought to solely infect mammals. The arenaviruses reported here share less than 15% amino-acid sequence similarity (in the RdRp) to those from mammals and snakes, and define a new monophyletic evolutionary group, the pescarenaviruses (*Figure 1A*). The absence of clear sequence homology in the glycoprotein, the difference in genome segmentation (*Shi et al., 2018*), as well as phylogenetic analysis of the replicase demonstrate that pescarenaviruses share a common but ancient ancestor with arenaviruses infecting snakes and mammals. We recommend these fish-infecting arenaviruses are assigned to the new genus *Pescarenavirus*, with those infecting Chinook and sockeye salmon being assigned to the species *Salmon pescarenavirus* (SPAV), strains 1 and 2, respectively.

Farmed Chinook salmon positive for SPAV-1 displayed pathology and symptoms consistent with disease including inflammation of the spleen and liver, as well as tubule necrosis and hyperplasia in the kidney. Clinically, salmon presented with yellow fluid on the pyloric caeca and swim bladder, pale gills with haemorrhaging on the surface, and anaemia. Wild Chinook and sockeye that tested positive for arenavirus infection, but which were clinically healthy when sampled, showed few histological lesions. In-situ hybridization revealed that arenaviruses were concentrated mainly in macrophage-like cells, melanomacrophages, red-blood cells (RBCs) and endotheliocytes (*Figure 3*). These findings are consistent with localisation of arenaviruses in mammals and snakes, although in contrast to snakes and fish, mammalian red blood cells are not nucleated so the similarity likely only extends to nucleated cells. SPAV-1 and −2 shared similar cell tropism within Chinook and sockeye salmon, respectively (*Figure 3—figure supplement 1*). In one out of the eight Chinook samples examined, moderate chronic-active hepatitis was reported, and staining for SPAV-1 was identified in the area affected by inflammation (*Figure 3C and D*), while in the other samples SPAV-1 was confined to reticuloendothelial cells in the liver tissue or in the sinusoids. More lesions were observed in dead farmed Chinook, where disease progression is more advanced. Our observations indicate that arenaviruses are replicating in red-blood cells, and occur in the macrophages and leukocytes that consume the infected cells. Moreover, the observed pathological changes in arenavirus-infected fish, including anaemia, and lesions in the gills, kidney and liver would be expected for viruses that infect red-blood cells. These results are the first empirical evidence for arenavirus infection in fish, and suggest that SPAV, like many other arenaviruses, has the potential to be a causative agent of disease.

Sequencing of cultured Chinook salmon also revealed a previously undescribed nidovirus and reovirus. Phylogenetic analysis of the reovirus, named *Chinook aquareovirus* (CAV), predicts that it is part of the genus, *Aquareovirus* (*Figure 1B*). Rather than being most closely related to known reoviruses of salmon (*Winton et al., 1981*), CAV groups with a growing number of aquareoviruses, some of which are known to cause haemorrhagic disease and have led to serious losses to aquaculture in China (*Nibert and Duncan, 2013*; *Wang et al., 2012*). The observed clinical signs (anemia, dark spleen, and blood-filled kidneys) in dead farmed Chinook salmon with high loads of CAV are consistent with a haemorrhagic manifestation.

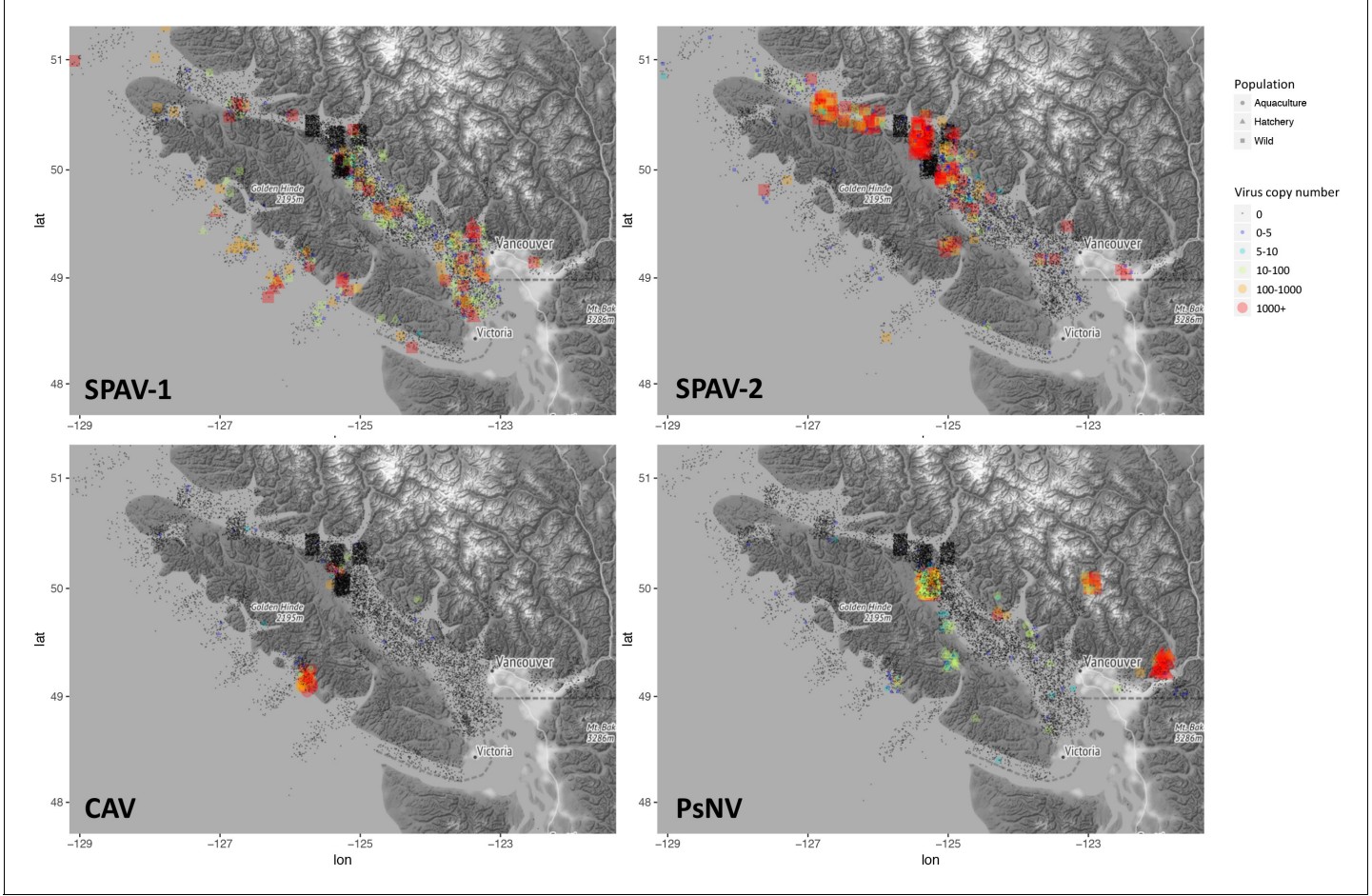

**Figure 2.** Epidemiological maps of Salmon pescarenavirus 1 and 2 (SPAV-1 and SPAV-2), Chinook aquareovirus (CAV) and Pacific salmon nidovirus (PsNV) around the coast of Vancouver Island. Individual samples are shown at the location collected, negative samples are black, and positive samples are coloured and sized according to the virus copy number. A small degree of random noise was added to the longitude and latitude to prevent overplotting.

DOI: https://doi.org/10.7554/eLife.47615.012

The following source data and figure supplements are available for figure 2:

**Source data 1.** Source data (RT-PCR copy number and sampling locations) for hte epidemiological maps.
DOI: https://doi.org/10.7554/eLife.47615.015
**Figure supplement 1.** Summary of RT-PCR for SPAV-1 and −2, PsNV and CAV using the Biomark Fluidigm platform.
DOI: https://doi.org/10.7554/eLife.47615.013
**Figure supplement 2.** Epidemiological maps from Washington to Alaska of Salmon pescarenavirus 1 and 2 (SPAV-1 and SPAV-2).
DOI: https://doi.org/10.7554/eLife.47615.014

The novel nidovirus, named *Pacific salmon nidovirus* (PsNV), is most closely related to the recently described Microhyla alphaletovirus 1, which forms a sister group to the coronaviruses (*Bukhari et al., 2018*). This sequence, alongside PsNV are basal to all other Nidovirus families, and their long branch length suggests they each belong to a different genus (*Figure 1C*). While not all coronaviruses cause serious disease, many do, such as SARS and MERS, which cause severe respiratory infections (*de Wit et al., 2016*).

Both SPAV-1 and SPAV-2 were relatively widespread along the coast of southwestern British Columbia, in ocean caught Chinook and sockeye salmon. Currently, it is unclear what is driving differences in SPAV-1 and SPAV-2 prevalence among regions, but the virus appears to be transmitted to juvenile salmon throughout southern BC soon after they enter the ocean, a period known to be critical to their survival (*Beamish et al., 2012a*). SPAV-1 was also relatively common in farmed Chinook populations. The distribution of SPAV-1 in wild Chinook populations was more localised to the

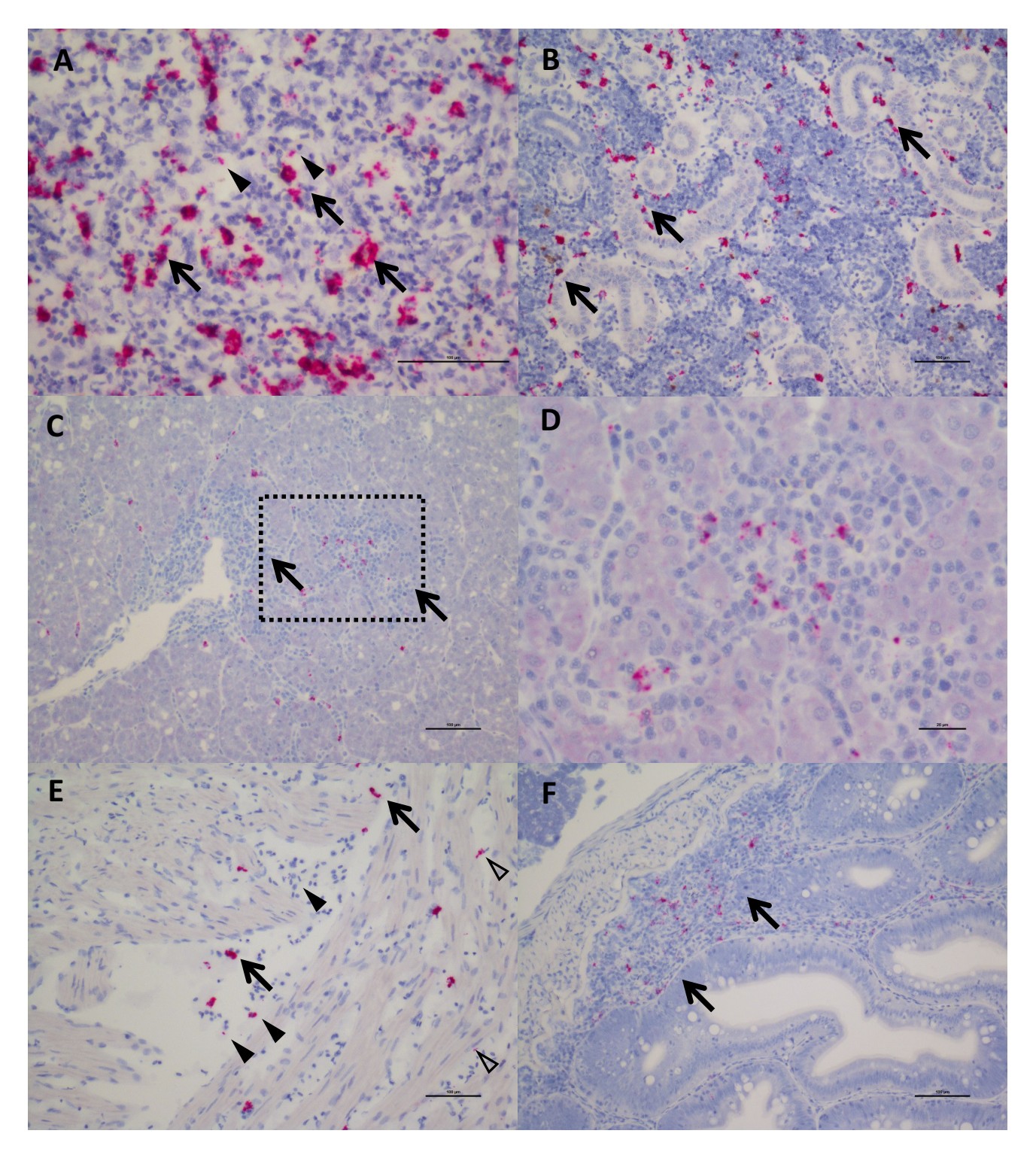

**Figure 3.** In Situ Hybridization staining of SPAV-1 in Chinook salmon. The red stain indicates localisation of viral RNA as well as viral transcripts. (**A**) Spleen: staining mostly localised in the macrophages (arrows) located around the sinusoids, although scattered positive red blood cells (arrowheads) are also present (scale bar 50 µm). (**B**) Posterior Kidney: the virus appears to be primarily localised in the peritubular capillaries (renal portal vessels) and macrophages (arrows) (scale bar 100 µm). (**C**) Liver: nodules of inflammation are mainly concentrated in a highly marked area. (scale bar 100 µm), dashed rectangle is enlarged in (**D**) showing lymphocytes and macrophages in the inflammatory nodule (several of which are positive for the virus).
*Figure 3 continued on next page*

*Figure 3 continued*

(scale bar 20 µm). (E) Heart: positive macrophages (arrows) are present between the fibres of the spongy myocardium, along with several positive red blood cells (arrowhead) and endothelial cells (open arrowheads). (scale bar 100 µm). (F) Intestine: staining for SPAV-1 is primarily localised to the gut-associated lymphoid tissue (arrows). (scale bar 100 µm).

DOI: https://doi.org/10.7554/eLife.47615.016

The following figure supplement is available for figure 3:

**Figure supplement 1.** In Situ Hybridization staining of SPAV-2 in sockeye salmon.

DOI: https://doi.org/10.7554/eLife.47615.017

west coast of Vancouver Island than SPAV-2, which was most prevalent on the east coast of Vancouver Island, near the Discovery Islands and the Johnstone Strait, and was rarely detected in sockeye salmon in northern BC and Alaska (*Figure 2—figure supplement 2*).

On the east coast of Vancouver Island, the Johnstone Strait and Discovery Islands have been identified as a potential choke point for the growth and survival of juvenile salmonids (*Healy et al., 2017*). The availability of prey to juvenile sockeye in the northern Johnstone Strait is extremely low, resulting in food limitation and increased competition for prey (*Beamish et al., 2012a*; *McKinnell et al., 2014*; *Godwin et al., 2015*; *Godwin et al., 2018*). These regions of high SPAV-2 infection could represent a stressful part of juvenile sockeye outmigration, possibly resulting in higher susceptibility to infection. Moreover, SPAV-2 was detected at high loads in fish sampled from regions where finfish aquaculture facilities are abundant and accordingly, sea lice infestation is high (*Price et al., 2011*]. It remains an open question whether an alternative host could play a role in virus transmission between fish, and/or result in an increased susceptibility to infection (*Valdes-Donoso et al., 2013*).

The distribution of CAV was markedly different from SPAV. CAV was not detected in any juvenile wild or hatchery Chinook salmon, despite being detected in farmed fish on both the west and east coasts of Vancouver Island. Over 20% of moribund Chinook aquaculture fish tested positive for CAV, with most detections occurring in fish at least 1.5 years after ocean entry, well past the time when migratory salmon were sampled. Hence, infection by CAV may take a considerable time to develop, or be an infection that is only acquired by older fish. CAV was also detected in a small number of farmed Atlantic salmon (seven positive detections of 2816 fish tested). The monophyletic grouping of CAV with other disease causing aquareoviruses and the consistency with haemorrhagic disease suggest that the virus is important to monitor in cultured fish, and potentially wild adults returning after several years at sea.

PsNV distribution was strongly associated with a handful of salmon-enhancement hatcheries but was also detected in 18% of aquaculture Chinook and 3% of wild Chinook (*Figure 2—figure supplement 1*). In hatchery fish, infection by PsNV was primarily localised to gill tissue (*Figure 4A*), reminiscent of the respiratory disease caused by the related mammalian coronaviruses such as MERS and SARS (*Figure 1C*). PsNV is of particular concern as it proliferates while fish are undergoing smoltification, a process during which the gill tissue undergoes cellular reconfiguration to prepare for saltwater. Notably, branchial proliferation of no known cause was noted in some farmed salmon infected with PsNV. In one of the hatcheries, where pre- and post-release sampling took place, the virus increased in prevalence during smolt development in fresh water, was detected shortly post-release, and was barely detected in the month following ocean entry (*Figure 4B*). This suggests that infected fish either cleared the infection, or did not survive after entry into the marine environment. The second interpretation is consistent with the lower rates of ocean survival in fish produced from hatcheries versus wild salmon (*Beamish et al., 2012b*).

Viral disease is a potential threat to wild fish stocks; yet little is known about viruses circulating in wild, farmed, or hatchery salmon. Here, through metatranscriptomic surveys, we reveal several previously unknown viruses that were discovered in dead and dying aquaculture fish, and show them to also occur in wild and hatchery-reared fish. Depending on the viral and host species, the viruses range from being localised to widespread, from infecting <1% to >20% of fish, and being from within the limits of detection to very high loads. Our results are consistent with some of these viruses being causative agents of disease, making it critical to understand their possible roles in salmon mortality and the decline of wild salmon populations, and their potential interactions with net-pen fish farming and hatchery rearing. Viral discovery in moribund individuals followed by extensive

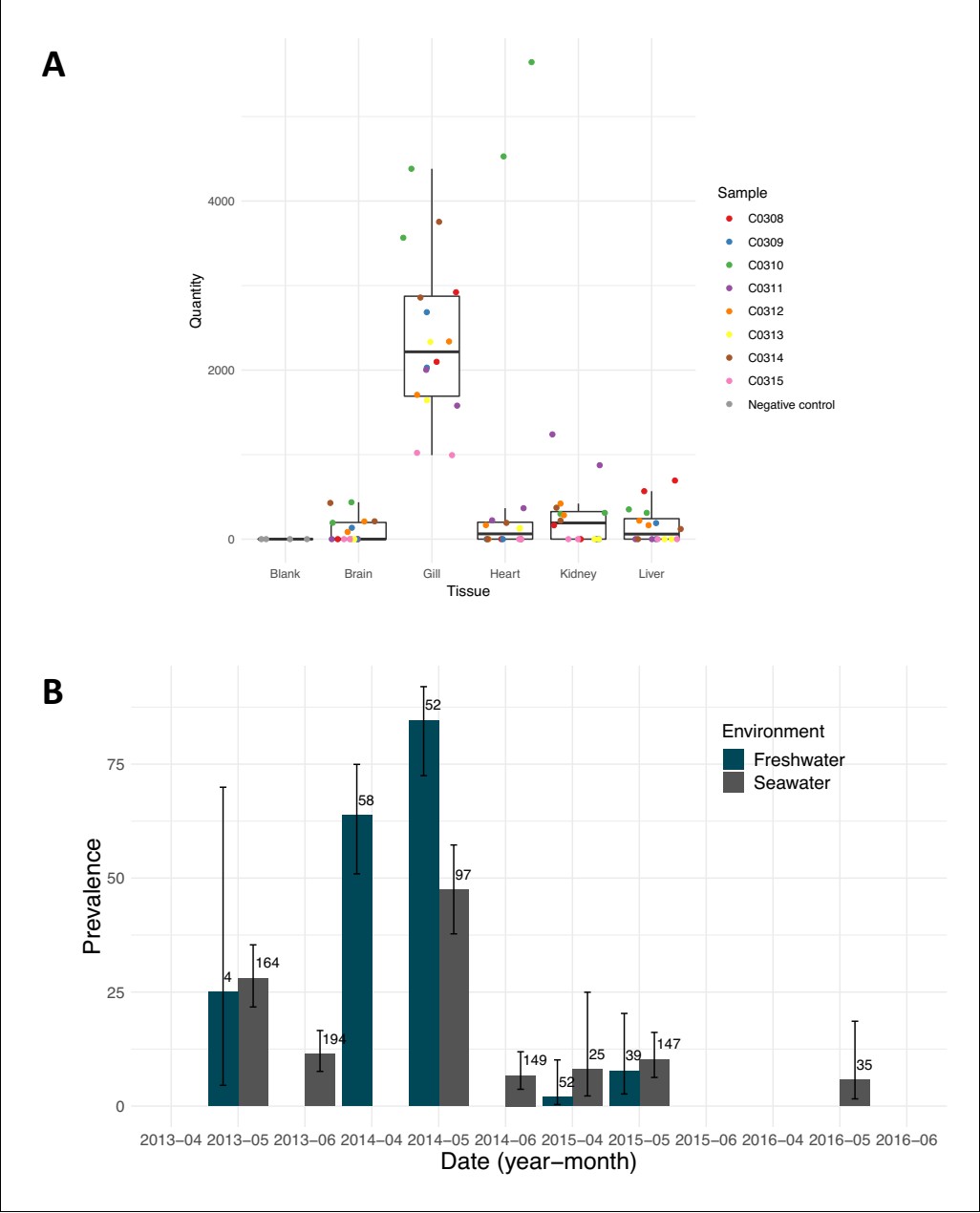

**Figure 4.** Pacific salmon nidovirus localisation and detections at a single salmon enhancement hatchery. (**A**) Average relative quantity of Pacific salmon nidovirus in dissected tissues of eight Chinook. Each sample was run and plotted in duplicate. (**B**) Prevalence of Pacific salmon nidovirus in fish collected in fresh and saltwater at a single salmon enhancement hatchery over four years. The data shown are the prevalence of positive amplifications above the calculated limit of detection (95%). Numbers show the sample size and error bars show Wilson's binomial confidence intervals.

DOI: https://doi.org/10.7554/eLife.47615.018

surveillance and histopathological localisation are powerful tools towards the ultimate goals of identifying causative agents of disease and understanding the impact of infectious agents in wild populations. These insights are crucial as juvenile salmon that are in less than optimal health are expected to have lower rates of survival in the wild. Continued surveillance and knowledge of endemic and emerging virus infections in these iconic salmon species is beneficial for their conservation.

## Materials and methods

### Nucleic acid extractions

Samples were provided by the Fisheries and Oceans, Canada Aquaculture Management Division and Salmon Enhancement Program. Additional samples were collected by the Hakai Institute Juvenile Salmon Program. Hatchery samples are identified by fin clipping, and in this study, wild fish could also encompass unmarked hatchery fish. DNA is extracted for detection of DNA viruses, bacteria and parasites from the same tissues from which we extract RNA to target RNA viruses. Nucleic acid extractions on the audit samples (eight tissues-gill, atrium, ventricle, liver, pyloric caeca, spleen, head kidney and posterior kidney) were as previously described (*Laurin et al., 2019*). For the wild Chinook and sockeye samples, homogenization using Tri-reagent was performed in a Mixer Mill (Qiagen, Maryland) on each tissue independently (five tissues- gill, liver, heart, head kidney and brain). Tri-reagent homogenates were organically separated using bromochloropropane, with the RNA-containing aqueous layer removed for RNA extraction and the lower DNA-containing organic layer separated from the organics using a TNES-Urea Buffer (*Asahida et al., 1996*).

For the DNA extractions, a pool of 250 µl (5 tissues contributing 50 µl each) from each of the tissue TNES aqueous layers was processed for DNA using the BioSprint 96 DNA Blood kit (Qiagen, Maryland) and the BioSprint 96 instrument (Qiagen, Maryland) both based on manufacturer's instructions. DNA was quantified using spectrophotometer readings performed on the Infinite M200Pro spectrophotometer (Tecan Group Ltd., Switzerland) and normalised to 62.5 ng/µl using the Freedom Evo (Tecan Group Ltd., Switzerland) liquid handling unit, based on manufacturer's instructions.

Similarly, a pool of 100 µl (5 tissues contributing 20 ul each) of the aqueous layer was processed for RNA using the Magmax−96 for Microarrays RNA kit (Ambion Inc, Austin, TX, USA) with a Biomek NXP (Beckman-Coulter, Mississauga, ON, Canada) automated liquid-handling instrument, both based on manufacturer's instructions. The quantity of RNA was analysed using spectrophotometer readings and normalised to 62.5 ng/µl with a Biomek NXP (Beckman-Coulter, Mississauga, ON, Canada) automated liquid-handling instrument, based on manufacturer's instructions. Mixed tissue RNA (1 µg) was reverse transcribed into cDNA using the superscript VILO master mix kit (Invitrogen, Carlsbad, CA), following the manufacturer's instructions.

### Metatranscriptomic sequencing

We applied a panel of host biomarkers (genes) that when co-expressed are indicative of a viral disease state (VDD) (*Miller et al., 2017*). Samples that displayed a positive viral disease state, but were not positive for viruses based on our 45 microbe panel screening, (as described in *Bass et al., 2019*), were selected for high throughput sequencing of RNA (dual RNA-seq) to discover new viral agents.

Total RNA from the mixed tissue samples was evaluated for quality using the Total RNA Pico chip on the Agilent 2100 Bioanalyzer (Agilent, Santa Clara, CA) and quantified using the Qubit RNA Br kit (Invitrogen, Carlsbad, CA). A 1/100 dilution of the ERCC RNA Spike-In control mix 1 (Ambion, Carlsbad, CA) was added to each total RNA sample prior to ribosomal depletion and library preparation. The sequencing libraries and ribosomal removal were performed using the Epicentre ScriptSeq Complete Gold Kit (Epidemiology) (Illumina, San Diego, CA) according to manufacturer's instructions and included a positive control (Universal Human Reference RNA) (Agilent, Santa Clara, CA) and negative control (no total RNA). The rRNA depleted total RNA was purified using the Zymo RNA Clean and Concentrate-5 kit (Zymo Research, Irvine, CA) according to manufacturer's instructions and quantified using the Qubit RNA HS kit (Invitrogen, Carlsbad, CA). The ScriptSeq Index reverse primers were added to the cDNA during the final amplification step which involved 14 cycles. The 3'-terminal tagged cDNA and final amplified library were purified using the Agencourt AMPure XP system (Beckman Coulter, Brea, CA). The final library size was determined using the HS DNA chip on the Agilent 2100 Bioanalyzer (Agilent, Santa Clara, CA) and the concentration was determined using the Qubit dsDNA HS kit (Invitrogen, Carlsbad, CA). Sample libraries were normalised to 4 nM, pooled appropriately and denatured and diluted to obtain a final library of 17pM. Prior to loading into a v3 2 × 300 bp kit (Illumina, San Diego, CA), 2% phiX was spiked in. Finally, a paired-end 251 bp sequencing run was performed on the Illumina MiSeq System (Illumina, San Diego, CA), with four samples barcoded and pooled for each run.

To sequence SPAV-2, PsNV and CAV, the samples were prepared using the same method as above but sequenced by BC Cancer Agency using a HiSeq (2 × 125) protocol (four different samples indexed over one lane).

## Sequence analysis

The quality of the raw reads was checked using FASTQC (v0.11.7) (https://www.bioinformatics.bab-raham.ac.uk/projects/fastqc/). Low quality reads or regions of adapter sequences were removed using Trimmomatic (v0.36) (*Bankevich et al., 2012*). Reads were aligned to the Atlantic Salmon genome using bwa mem (v0.7.17-r1188) and unmapped reads were retained. The unmapped reads were then balanced using Trimmomatic and assembled into contigs using SPAdes (v3.9.1) genome assembler (*Bankevich et al., 2012*). Putative viral contigs were identified by aligning translated contigs using DIAMOND (v0.9.16.117) (*Buchfink et al., 2015*) to the nr database. Reference alignments of all the reads to the viral contigs were used to ensure that no assembly artefacts occurred and the contigs were trimmed appropriately using Geneious (V10.1.3). Assembled sequences are available on Genbank (BioProject: PRJNA547678, Genbank accession numbers: MK611979 - MK611996) and raw sequencing reads have been uploaded to the Sequence Read Archive (SAMN11974798 - SAMN11974801).

## Phylogenetic analysis

The phylogeny of each virus was resolved based on the predicted replicase (CAV and SPAV) and ORF1ab (PsNV) amino acid sequences, as nucleotide sequences were too dissimilar to reliably align. Alignments were generated with MAFTT (v7.42) (*Katoh and Standley, 2013*) employing the E-INS-i algorithm. This alignment algorithm is suited for evolutionarily distinct sequences with conserved motifs (such as viral RNA polymerase) that are embedded within long unalignable residues. The novel salmon viruses were aligned with other viral genomes with shared amino acid similarity as detected by DIAMOND (*Buchfink et al., 2015*). In addition, viral genomes which are known to be evolutionarily related to these were included. The multiple protein alignments were then used as to build phylogenies using PhyML 3.0 (*Guindon et al., 2010*) plugin within Geneious with 100 boot-straps to generate branch support values. Trees are mid-point rooted for clarity only, and do not necessarily represent the ancestral relationship of the viruses.

## Assay development and screening

Assembled viral sequence contigs from the appropriate sample were imported into Primer Express v3.0.1 software (Thermo Fisher Scientific, Waltham, MA) where qPCR Taqman assays were designed using default parameters (*Supplementary file 1*). These assays were then tested using the Fluidigm BioMark microfluidics-based qPCR system following the same protocol as described below except with the new viral primer pairs included in the STA step and controls. From these initial screens, the most consistent assay was chosen and APC standards were constructed to include in future Fluidigm BioMark qPCR microbe panels. The assay-specific theoretical limit of detection was calculated as previously described (*Miller et al., 2016*). The limit of detection was applied to categorise fish with amplifications above the 95% detection threshold that is the concentration of the analyte in the sample matrix that would be detected with high statistical certainty (95% of the time). Epidemiological maps were generated using these data with the limit of detection applied. The maps were created within R using ggplot2 (*Wickham, 2016*) and ggmap (*Kahle and Wickham, 2013*).

## RT-PCR

For all samples, after reverse transcription, resultant cDNA was combined with the normalised DNA in a ratio of 1:1 and used as the template for the specific target amplification (STA) step. The STA involves a pre-amplification of all primers to be run on a single dynamic array at low concentrations (0.2 µM of each of the primers), and upon completion, excess primers were removed by treating with Exo-SAP-IT (Affymetrix, Santa Clara, CA) according to manufacturer's instructions and then diluted 1:5 in DNA re- suspension buffer (Teknova, Hollister, CA).

The 96.96 gene expression dynamic array (Fluidigm Corporation, CA, US) was run according to the procedure outlined previously (*Miller et al., 2016*). Specifically, a 5 µl template mixture was

prepared for each sample containing 1 × TaqMan Universal Master Mix (No UNG), 1 × GE Sample Loading Reagent (Fluidigm PN 85000746) and each of diluted STA'd sample mixtures. Five μl of Assay mix was prepared with 1 × each of the appropriate TaqMan qPCR assays (agent probe in FAM-MGB and artificial positive construct (APC) probe in NED-MGB, 10 μM of primers and 3 μM of probes) and 1 × Assay Loading Reagent (Fluidigm PN 85000736).

Controls were added prior to running the dynamic array (*Miller et al., 2016*). Note, APC clones to all assays were contained in a single serially diluted pool, loaded last, minimising the likelihood of contamination of any single APC clone. Once loading and mixing of the dynamic array was completed within the IFC HX controller, the array was transferred to the BioMark HD instrument and processed using the GE 96 × 96 Standard TaqMan program for qPCR which includes a hot start followed by 40 cycles at 95℃ for 15 s and 60℃ for 1 min (Fluidigm Corporation, CA, USA). The data were analysed with Real-Time PCR Analysis Software (Fluidigm Corporation, CA, USA).

Chinook smolt samples positive for PsNV from 2014 were used for tissue localization (*Figure 4A*). Gill, liver, heart, kidney, and brain were individually homogenized, processed for RNA extraction (as described above), and 1 ug normalised RNA was used for reverse transcription. Resultant cDNA for each individual tissue was used as the template for PsNV relative quantification using an ABI 7900HT (ABI) in 384-well optical plates. The qPCR reaction volume was 12 μl, which comprised 6 μl of 2X TaqMan Gene Expression Master Mix (ABI PN 4369016), 4.3 μl of water, 0.22 μl of mixed forward and reverse primers (900 nM final concentration of each), 0.24 μl of each probe (200 nM final concentration; assay specific probe and APC control probe), and 1 μl of cDNA template. Temperature cycles included one 2 min hold (50℃), a 10 min denaturation (95℃), and 40 cycles of denaturation (95℃ for 15 s), annealing and extension (60℃ for 60 s). Amplification conditions on the ABI 7900 were not optimised for this platform, but rather closely reflected those used on the BioMark platform. Samples run on the ABI did not undergo STA enrichment. Standard curves were constructed using the same APC clone standards spiked in with CHSE DNA as on the BioMark. Serial dilutions were made to obtain concentrations of 24, $1.2 \times 10^2$, $6 \times 10^2$, $3 \times 10^3$, $1.5 \times 10^4$, $1.5 \times 10^5$ copies of the clone per reaction. Clone standards, unknown samples, positive and negative controls were all run in duplicate. The ABI software calculates the relative copy number based upon the serial dilution of the standard curve.

## Histopathology

Before the discovery of these viruses clinical signs of disease and histopathological lesions were assessed for approximately 230 farmed Chinook salmon sampled in the Audit program. Consequently, gills, skeletal muscle, spleen, liver, heart, anterior and posterior kidney, pyloric caeca and brain from eleven samples of Chinook (eight wild fish and three farmed fish) and ten sockeye (all wild fish) positive for SPAV were histopathologically analysed to assess the presence of lesions. All tissues were fixed in 10% neutral buffered formalin, dehydrated through an ascending gradient of alcohol solutions, embedded in paraffin wax, cut at 3.5 μm thickness, and stained with routine hematoxylin and eosin (H and E) for morphological evaluation by light microscope.

## In Situ Hybridization (ISH)

RNA-ISH was performed using RNAscope 2.5 HD Duplex assay (Advanced Cell Diagnostics, Newark, California, USA, catalog# 322500) according to the manufacturer's instructions. Briefly, consecutive sections of Chinook and sockeye salmon samples utilised for the histopathological analysis were dewaxed by incubating for 60 min at 60℃ and endogenous peroxidases were quenched with hydrogen peroxide for 10 min at room temperature. Slides were then boiled for 30 min in RNAscope target retrieval reagents (Advanced Cell Diagnostics, Newark, California, USA) and incubated for 30 min in RNAscope Protease Plus reagent prior to hybridization. The slides underwent hybridization with RNAscope probes against a portion of SPAV-1 and SPAV-2 genome (Advanced Cell Diagnostics, Newark, California, USA, catalog #513591-C2 and 538881-C2, respectively). A RNAscope probe against Coil-p84 housekeeping gene in Chinook salmon (Advanced Cell Diagnostics, Newark, California, USA, catalog #512391) was used as positive control probe to confirm the efficacy of the probes and the viability of the samples. Two samples which were negative for SPAV-1 and SPAV-2 were used as negative controls to confirm absence of background and (or) non-specific cross-

reactivity of the assay. Signal amplification was performed according to the manufacturer's instructions, followed by counterstaining with Gill's hematoxylin and visualisation by bright field microscopy.

## Acknowledgements

We thank Jeffrey Joy, Veronica Paglowski, Jan Finke, Thomas Waltzek, Ben Neuman and Humberto Debat for advice and help with the analysis, DFO Aquaculture Management Division, Salmon Enhancement Program and the Hakai Institute Juvenile Salmon Program for provision of samples, and all of the vessel and field crews who collected the thousands of samples. We thank Amy Chan and all members of the Suttle lab for their feedback as the project developed. This research was supported by funding for the Strategic Salmon Health Initiative (SSHI), which is part of the Salish Sea Marine Survival Project. The SSHI seeks to discover microbes present in salmon in British Columbia that may be undermining the productivity of Pacific salmon, and is funded by the Pacific Salmon Foundation (PSF), Genome British Columbia, and Fisheries and Oceans, Canada. This is publication number 32 from the Salish Sea Marine Survival Project (marinesurvivalproject.com).

## Additional information

### Funding

| Funder | Grant reference number | Author |
|---|---|---|
| Pacific Salmon Foundation | SSHI | Gideon J Mordecai<br>Kristina M Miller<br>Emiliano DiCicco<br>Angela D Schulze<br>Karia H Kaukinen<br>Tobi J Ming<br>Shaorong Li<br>Amy Tabata<br>Amy Teffer |
| Genome British Columbia | SSHI | Gideon J Mordecai<br>Kristina M Miller<br>Emiliano DiCicco<br>Angela D Schulze<br>Karia H Kaukinen<br>Tobi J Ming<br>Shaorong Li<br>Amy Tabata<br>Amy Teffer |
| Mitacs | SSHI | Gideon J Mordecai |

The funders had no role in study design, data collection and interpretation, or the decision to submit the work for publication.

### Author contributions

Gideon J Mordecai, Conceptualization, Formal analysis, Investigation, Visualization, Methodology, Writing—original draft, Writing—review and editing, Led the analyses and prepared the figures; Kristina M Miller, Conceptualization, Resources, Supervision, Funding acquisition, Investigation, Project administration, Writing—review and editing, Conceived the study; Emiliano Di Cicco, Formal analysis, Investigation, Writing—original draft, Conducted pathology, histopathology and in situ hybridisation; Angela D Schulze, Investigation, Methodology, Performed sequencing, nucleic acid extractions and molecular analyses; Karia H Kaukinen, Investigation, Methodology, Conducted field sampling, nucleic acid extractions and molecular analyses including tissue localisation by RT-PCR; Tobi J Ming, Investigation, Methodology, Conducted field sampling, nucleic acid extractions and molecular analyses; Shaorong Li, Data curation, Investigation, Methodology, Conducted nucleic acid extractions and molecular analyses; Amy Tabata, Data curation, Conducted field sampling, nucleic acid extractions and molecular analyses; Amy Teffer, Investigation, Writing—review and editing, Conducted field sampling, nucleic acid extractions and molecular analyses; David A Patterson,

Conceptualization, Investigation, Writing—review and editing, Conducted field sampling, nucleic acid extractions and molecular analyses; Hugh W Ferguson, Investigation, Writing—review and editing, Conducted pathology, histopathology and in situ hybridisation; Curtis A Suttle, Resources, Supervision, Writing—review and editing, Conceived the study

### Author ORCIDs
Gideon J Mordecai (iD) https://orcid.org/0000-0001-8397-9194
Curtis A Suttle (iD) https://orcid.org/0000-0002-0372-0033

### Decision letter and Author response
Decision letter https://doi.org/10.7554/eLife.47615.061
Author response https://doi.org/10.7554/eLife.47615.062

## Additional files

### Supplementary files
• Supplementary file 1. Table of primers and taqman assays used in this study.
DOI: https://doi.org/10.7554/eLife.47615.019

• Supplementary file 2. Commands used in the bioinformatic pipeline.
DOI: https://doi.org/10.7554/eLife.47615.020

• Transparent reporting form
DOI: https://doi.org/10.7554/eLife.47615.021

### Data availability
Assembled viral genomes have been deposited to Genbank under accession numbers MK611979–MK611996 and sequencing reads have been submitted to the Sequence Read Archive under the accession: PRJNA547678.

The following datasets were generated:

| Author(s) | Year | Dataset title | Dataset URL | Database and Identifier |
|-----------|------|---------------|-------------|-------------------------|
| Gideon J Mordecai, Kristina M Miller, Emiliano Di Cicco, Angela D Schulze, Karia H Kaukinen, Tobi J Ming, Shaorong Li, Amy Tabata, Amy Teffer, David A Patterson, Hugh W Ferguson, Curtis A Suttle | 2019 | Endangered wild salmon infected by newly discovered viruses | https://www.ncbi.nlm.nih.gov/sra/PRJNA547678 | NCBI Sequence Read Archive, PRJNA547678 |
| Gideon J Mordecai, Kristina M Miller, Emiliano Di Cicco, Angela D Schulze, Karia H Kaukinen, Tobi J Ming, Shaorong Li, Amy Tabata, Amy Teffer, David A Patterson, Hugh W Ferguson, Curtis A Suttle | 2019 | Endangered wild salmon infected by newly discovered viruses | https://www.ncbi.nlm.nih.gov/nuccore/?term=MK611979 | NCBI Genbank, MK611979 |
| Gideon J Mordecai, Kristina M Miller, Emiliano Di Cicco, Angela D Schulze, Karia H Kaukinen, Tobi J Ming, Shaorong Li, Amy Tabata, Amy Teffer, David A Patterson, | 2019 | Endangered wild salmon infected by newly discovered viruses | https://www.ncbi.nlm.nih.gov/nuccore/?term=MK611980 | NCBI Genbank, MK611980 |

| | | | | |
|---|---|---|---|---|
| Hugh W Ferguson, Curtis A Suttle | | | | |
| Gideon J Mordecai, Kristina M Miller, Emiliano Di Cicco, Angela D Schulze, Karia H Kaukinen, Tobi J Ming, Shaorong Li, Amy Tabata, Amy Teffer, David A Patterson, Hugh W Ferguson, Curtis A Suttle | 2019 | Endangered wild salmon infected by newly discovered viruses | https://www.ncbi.nlm.nih.gov/nuccore/?term=MK611981 | NCBI Genbank, MK611981 |
| Gideon J Mordecai, Kristina M Miller, Emiliano Di Cicco, Angela D Schulze, Karia H Kaukinen, Tobi J Ming, Shaorong Li, Amy Tabata, Amy Teffer, David A Patterson, Hugh W Ferguson, Curtis A Suttle | 2019 | Endangered wild salmon infected by newly discovered viruses | https://www.ncbi.nlm.nih.gov/nuccore/?term=MK611982 | NCBI Genbank, MK611982 |
| Gideon J Mordecai, Kristina M Miller, Emiliano Di Cicco, Angela D Schulze, Karia H Kaukinen, Tobi J Ming, Shaorong Li, Amy Tabata, Amy Teffer, David A Patterson, Hugh W Ferguson, Curtis A Suttle | 2019 | Endangered wild salmon infected by newly discovered viruses | https://www.ncbi.nlm.nih.gov/nuccore/?term=MK611983 | NCBI Genbank, MK611983 |
| Gideon J Mordecai, Kristina M Miller, Emiliano Di Cicco, Angela D Schulze, Karia H Kaukinen, Tobi J Ming, Shaorong Li, Amy Tabata, Amy Teffer, David A Patterson, Hugh W Ferguson, Curtis A Suttle | 2019 | Endangered wild salmon infected by newly discovered viruses | https://www.ncbi.nlm.nih.gov/nuccore/?term=MK611984 | NCBI Genbank, MK611984 |
| Gideon J Mordecai, Kristina M Miller, Emiliano Di Cicco, Angela D Schulze, Karia H Kaukinen, Tobi J Ming, Shaorong Li, Amy Tabata, Amy Teffer, David A Patterson, Hugh W Ferguson, Curtis A Suttle | 2019 | Endangered wild salmon infected by newly discovered viruses | https://www.ncbi.nlm.nih.gov/nuccore/?term=MK611985 | NCBI Genbank, MK611985 |
| Gideon J Mordecai, Kristina M Miller, Emiliano Di Cicco, Angela D Schulze, Karia H Kaukinen, Tobi J Ming, Shaorong Li, Amy Tabata, Amy Teffer, David A Patterson, Hugh W Ferguson, Curtis A Suttle | 2019 | Endangered wild salmon infected by newly discovered viruses | https://www.ncbi.nlm.nih.gov/nuccore/?term=MK611986 | NCBI Genbank, MK611986 |
| Gideon J Mordecai, Kristina M Miller, | 2019 | Endangered wild salmon infected by newly discovered viruses | https://www.ncbi.nlm.nih.gov/nuccore/?term= | NCBI Genbank, MK611987 |

| | | | | |
|---|---|---|---|---|
| Emiliano Di Cicco, Angela D Schulze, Karia H Kaukinen, Tobi J Ming, Shaorong Li, Amy Tabata, Amy Teffer, David A Patterson, Hugh W Ferguson, Curtis A Suttle | | | MK611987 | |
| Gideon J Mordecai, Kristina M Miller, Emiliano Di Cicco, Angela D Schulze, Karia H Kaukinen, Tobi J Ming, Shaorong Li, Amy Tabata, Amy Teffer, David A Patterson, Hugh W Ferguson, Curtis A Suttle | 2019 | Endangered wild salmon infected by newly discovered viruses | https://www.ncbi.nlm.nih.gov/nuccore/?term=MK611988 | NCBI Genbank, MK611988 |
| Gideon J Mordecai, Kristina M Miller, Emiliano Di Cicco, Angela D Schulze, Karia H Kaukinen, Tobi J Ming, Shaorong Li, Amy Tabata, Amy Teffer, David A Patterson, Hugh W Ferguson, Curtis A Suttle | 2019 | Endangered wild salmon infected by newly discovered viruses | https://www.ncbi.nlm.nih.gov/nuccore/?term=MK611989 | NCBI Genbank, MK611989 |
| Gideon J Mordecai, Kristina M Miller, Emiliano Di Cicco, Angela D Schulze, Karia H Kaukinen, Tobi J Ming, Shaorong Li, Amy Tabata, Amy Teffer, David A Patterson, Hugh W Ferguson, Curtis A Suttle | 2019 | Endangered wild salmon infected by newly discovered viruses | https://www.ncbi.nlm.nih.gov/nuccore/?term=MK611990 | NCBI Genbank, MK611990 |
| Gideon J Mordecai, Kristina M Miller, Emiliano Di Cicco, Angela D Schulze, Karia H Kaukinen, Tobi J Ming, Shaorong Li, Amy Tabata, Amy Teffer, David A Patterson, Hugh W Ferguson, Curtis A Suttle | 2019 | Endangered wild salmon infected by newly discovered viruses | https://www.ncbi.nlm.nih.gov/nuccore/?term=MK611991 | NCBI Genbank, MK611991 |
| Gideon J Mordecai, Kristina M Miller, Emiliano Di Cicco, Angela D Schulze, Karia H Kaukinen, Tobi J Ming, Shaorong Li, Amy Tabata, Amy Teffer, David A Patterson, Hugh W Ferguson, Curtis A Suttle | 2019 | Endangered wild salmon infected by newly discovered viruses | https://www.ncbi.nlm.nih.gov/nuccore/?term=MK611992 | NCBI Genbank, MK611992 |
| Gideon J Mordecai, Kristina M Miller, Emiliano Di Cicco, Angela D Schulze, Karia H Kaukinen, Tobi J Ming, Shaorong Li, Amy | 2019 | Endangered wild salmon infected by newly discovered viruses | https://www.ncbi.nlm.nih.gov/nuccore/?term=MK611993 | NCBI Genbank, MK611993 |

| | | | | |
|---|---|---|---|---|
| Tabata, Amy Teffer, David A Patterson, Hugh W Ferguson, Curtis A Suttle | | | | |
| Gideon J Mordecai, Kristina M Miller, Emiliano Di Cicco, Angela D Schulze, Karia H Kaukinen, Tobi J Ming, Shaorong Li, Amy Tabata, Amy Teffer, David A Patterson, Hugh W Ferguson, Curtis A Suttle | 2019 | Endangered wild salmon infected by newly discovered viruses | https://www.ncbi.nlm.nih.gov/nuccore/?term=MK611994 | NCBI Genbank, MK611994 |
| Gideon J Mordecai, Kristina M Miller, Emiliano Di Cicco, Angela D Schulze, Karia H Kaukinen, Tobi J Ming, Shaorong Li, Amy Tabata, Amy Teffer, David A Patterson, Hugh W Ferguson, Curtis A Suttle | 2019 | Endangered wild salmon infected by newly discovered viruses | https://www.ncbi.nlm.nih.gov/nuccore/?term=MK611995 | NCBI Genbank, MK611995 |
| Gideon J Mordecai, Kristina M Miller, Emiliano Di Cicco, Angela D Schulze, Karia H Kaukinen, Tobi J Ming, Shaorong Li, Amy Tabata, Amy Teffer, David A Patterson, Hugh W Ferguson, Curtis A Suttle | 2019 | Endangered wild salmon infected by newly discovered viruses | https://www.ncbi.nlm.nih.gov/nuccore/?term=MK611996 | NCBI Genbank, MK611996 |

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
