## [Decision Letter]

Thank you for submitting your article "Endangered wild salmon infected by newly discovered viruses" for consideration by *eLife*. Your article has been reviewed by three peer reviewers, including Richard A Neher as the Reviewing Editor and Reviewer #1, and the evaluation has been overseen by Ian Baldwin as the Senior Editor.

The reviewers have discussed the reviews with one another and the Reviewing Editor has drafted this decision to help you prepare a revised submission.

Summary:

Mordecai and colleagues identify viruses by metatranscriptomics in dead or diseased salmon, localize these viruses in host cells using in-situ hybridization and survey the prevalence of these viruses in farmed and wild salmon. They show that the combination of meta-genomic virus discovery with broad surveillance is an effective strategy to identify a pathogen and map its distribution. This is a potentially widely applicable strategy which makes this manuscript an interesting contribution. There are, however, a number of issues that need to be addressed.

Essential revisions:

1) We identified multiple issues concerning data sharing. The sequencing reads need to be uploaded to a sequence read archive and accession numbers need to be provided. All assembled contigs that map to RNA viruses should be provided as supplementary information. Alignments underlying the phylogenetic trees in Figure 1 need to be provided as fasta files, one file per alignment (not as a word file!). Furthermore, the phylogenetic trees should be provided as newick or nexus files as source data for Figure 1. Primers and probes should be given as supplementary material. Data not shown is not acceptable. If the point made is important, support it by data. Otherwise cut. Data underlying Figure 2 should also be provided.

2) Materials and methods need to contain more detail and should be edited for clarity and consistency. The description of the sequence analysis should contain parameters and version numbers of the software used. Ideally, scripts containing the explicit commands of the pipeline should be provided. The molecular methods are partly confusing or incomplete. The RT-PCR section, for example, describes DNA prep before reverse transcription. Names of kits, for example the BioSprint 96 kit, are not specific enough as there are many of BioSprint 96 kits for different applications. Similar issues apply elsewhere. Critical bits necessary to understand the paper (for example the provenance of the samples, hatchery/non-hatchery ambiguity) should be mentioned briefly in the main text. Other aspects need elaboration: How were sequences in the alignments underlying the phylogenetic trees selected? Alignments contain all-gap/X columns? Why are they not removed?

3) The choice of midpoint rooting is sub-optimal. Instead, rooting should reflect the current understanding of RNA virus evolution. For reoviruses the midpoint root appears to correspond to the orthoreovirus-aquareovirus split as intended. For arenaviruses the current view is that Haartman Institute snake virus forms an outgroup to the amniote-infecting arenaviruses and thus the clade of fish arenaviruses should be the outgroup. Likewise, for nidoviruses, the currently accepted view is that coronaviruses and toroviruses are more closely related to each other with arteriviruses as the outgroup. If the authors would prefer not to take a stand on what the correct root for each tree is they are free to display the trees as unrooted. Figure 1 could be made more useful by coloring different viral clades or by highlighting host species.

4) The authors call the two newly discovered arenaviruses different species (1 and 2) and go as far as identifying them as a new clade called Pescarenavirus. The group that the authors designate as Pescarenavirus also contains the Wenling frogfish arenaviruses described by Shi et al. We suggest to reach out the authors of this study and to agree on a name for this clade. The SPAV-1/2 viruses are very similar to each other and could also be simply two strains rather than species. If 20 new arenaviruses were discovered in salmon that fell directly within the diversity encompassed by the two lineages described here, would the authors really argue that they should be called Salmon pescarenavirus 3 through 23?

5) Some discussion of the genome organization of the newly discovered viruses would be valuable. At least comparison of the identified contigs to the closest BLAST hits and a discussion of how well these genomes/contigs are covered by the reads from the meta-transcriptomic sequencing.

---

## [Author Response]

Essential revisions:1) We identified multiple issues concerning data sharing. The sequencing reads need to be uploaded to a sequence read archive and accession numbers need to be provided.

Sequencing reads have now been uploaded to the SRA and accession numbers have been added to the Materials and methods, and is noted in the manuscript as follows:

“Assembled sequences are available on Genbank (Genbank accession numbers: MK611979 – MK611996) and raw sequencing reads have been uploaded to the Sequence Read Archive (SAMN11974798 – SAMN11974801).”

All assembled contigs that map to RNA viruses should be provided as supplementary information.

Fasta files of assembled viral contigs are now supplied (Figure 1—figure supplement 1—source data 1).

Alignments underlying the phylogenetic trees in Figure 1 need to be provided as fasta files, one file per alignment (not as a word file!).

Amino acid alignments have been provided as three separate fasta files (Figure

1—source data 1-3).

Furthermore, the phylogenetic trees should be provided as newick or nexus files as source data for Figure 1.

These are now provided as source data – Figure 1—source data 4-6.

Primers and probes should be given as supplementary material. Data not shown is not acceptable. If the point made is important, support it by data. Otherwise cut. Data underlying Figure 2 should also be provided.

Underlying data for Figure 2 is now supplied (Figure 2—source data 1). Primer and probe sequences have been provided as Supplementary file 1, which is cited in the text in the ‘Assay development and screening’ section of the ‘Materials and methods’. ‘Data not shown’ has been removed and the following details have been added “CAV was also detected in a small number of farmed Atlantic salmon (7 positive detections of 2816 fish tested).”

2) Materials and methods need to contain more detail and should be edited for clarity and consistency. The description of the sequence analysis should contain parameters and version numbers of the software used. Ideally, scripts containing the explicit commands of the pipeline should be provided.

Version numbers of the software have now been included within the text in the ‘Materials and methods’ section, and the command line scripts for the genome assembly have been included as Supplementary file 2.

The molecular methods are partly confusing or incomplete. The RT-PCR section, for example, describes DNA prep before reverse transcription.

We have reordered this section, and have added more explanation of why we also describe DNA extraction (subsection “Nucleic acid extractions”, first paragraph).

Names of kits, for example the BioSprint 96 kit, are not specific enough as there are many of BioSprint 96 kits for different applications.

We added more detail for the BioSprint 96 DNA Blood kit (Qiagen, Maryland) (subsection “Nucleic acid extractions”, second paragraph).

Similar issues apply elsewhere. Critical bits necessary to understand the paper (for example the provenance of the samples, hatchery/non-hatchery ambiguity) should be mentioned briefly in the main text.

This issue of hatchery/non-hatchery ambiguity is now mentioned in the Materials and methods (subsection “Nucleic acid extractions”, first paragraph) as well as in the figure legend for Figure 2.

Other aspects need elaboration: How were sequences in the alignments underlying the phylogenetic trees selected?

We have added more detail in the Materials and methods about how the sequences included in the phylogenies were selected: “The novel salmon viruses were aligned with other viral genomes with shared amino acid similarity as detected by DIAMOND. In addition, viral genomes which are known to be evolutionarily related to these were included.”

Alignments contain all-gap/X columns? Why are they not removed?

An explanation of the alignment algorithm used and why we think it is suitable has been added.

“This alignment algorithm is suited for evolutionarily distinct sequences with conserved motifs (such as viral RNA polymerase) which are embedded within long unalignable residues.” Within one of the alignments there were columns containing only gaps, and these have been removed.

3) The choice of midpoint rooting is sub-optimal. Instead, rooting should reflect the current understanding of RNA virus evolution. For reoviruses the midpoint root appears to correspond to the orthoreovirus-aquareovirus split as intended. For arenaviruses the current view is that Haartman Institute snake virus forms an outgroup to the amniote-infecting arenaviruses and thus the clade of fish arenaviruses should be the outgroup. Likewise, for nidoviruses, the currently accepted view is that coronaviruses and toroviruses are more closely related to each other with arteriviruses as the outgroup. If the authors would prefer not to take a stand on what the correct root for each tree is they are free to display the trees as unrooted.

Because RNA viral taxonomic groupings are often unclear, and subject to change, we prefer not to manually select the root. For instance, some believe that since the discovery of several new viruses in this group, the shallow Ortho/ Aquareovirus phylogenetic split is not large enough to warrant separate genera (see Nibert and Duncan, 2013). As a compromise, Tempest (http://tree.bio.ed.ac.uk/software/tempest/) was used to calculate the best fitting root for each tree, which should also help make the results more reproducible by others.

Figure 1 could be made more useful by coloring different viral clades or by highlighting host species.

Host groups are now shown using colour (Figure 1).

4) The authors call the two newly discovered arenaviruses different species (1 and 2) and go as far as identifying them as a new clade called Pescarenavirus. The group that the authors designate as Pescarenavirus also contains the Wenling frogfish arenaviruses described by Shi et al. We suggest to reach out the authors of this study and to agree on a name for this clade. The SPAV-1/2 viruses are very similar to each other and could also be simply two strains rather than species. If 20 new arenaviruses were discovered in salmon that fell directly within the diversity encompassed by the two lineages described here, would the authors really argue that they should be called Salmon pescarenavirus 3 through 23?

Agreed, SPAV-1 and -2 appear to be two different strains of the same species. Throughout the text, we refer to SPAV-1 and -2 as two different variants or strains of the same species ‘Salmon pescarenavirus’ which we believe form a new genus (along with other fish arenaviruses), the pescarenaviruses, which follows the naming convention already in place (i.e. mamma- and repta- arenaviruses).

For example, the Abstract states that we “revealed two different arenavirus variants which each infect wild Chinook and sockeye salmon” and: “…the species Salmon pescarenavirus, Strains SPAV-1 and SPAV-2, respectively”. We plan to submit a taxonomic proposal to ICTV to name the genus pescarenavirus, and will consult the authors of Shi et al.

5) Some discussion of the genome organization of the newly discovered viruses would be valuable. At least comparison of the identified contigs to the closest BLAST hits and a discussion of how well these genomes/contigs are covered by the reads from the meta-transcriptomic sequencing.

A new figure supplement for Figure 1 shows the genome organisation and ORF prediction for each virus; it is also mentioned in the text. Additionally, in the same figure coverage plots for all genomic segments are provided and emphasized as follows:

“The genomic organisation of the newly discovered viruses was consistent with related viruses in fish; for instance, SPAV has three genomic segments, as shown for other arenaviruses in fish”.